# Compatible Consortium of Endophytic *Bacillus halotolerans* Strains Cal.l.30 and Cal.f.4 Promotes Plant Growth and Induces Systemic Resistance against *Botrytis cinerea*

**DOI:** 10.3390/biology12060779

**Published:** 2023-05-27

**Authors:** Polina C. Tsalgatidou, Eirini-Evangelia Thomloudi, Costas Delis, Kallimachos Nifakos, Antonios Zambounis, Anastasia Venieraki, Panagiotis Katinakis

**Affiliations:** 1Laboratory of General and Agricultural Microbiology, Agricultural University of Athens, Iera Odos 75, 11855 Athens, Greece; polina.tsalgatidou@go.uop.gr (P.C.T.); e.e.thomloudi@gmail.com (E.-E.T.); 2Department of Agriculture, University of the Peloponnese, 24100 Kalamata, Greece; k.delis@uop.gr (C.D.); k.nifakos@uop.gr (K.N.); 3Institute of Plant Breeding and Genetic Resources, Hellenic Agricultural Organization ‘ELGO DIMITRA’, 57001 Thessaloniki, Greece; azampounis@elgo.gr; 4Laboratory of Plant Pathology, Agricultural University of Athens, Iera Odos 75, 11855 Athens, Greece

**Keywords:** *Bacillus halotolerans*, seed bio-priming, soil drenching, biofertilizers, biocontrol agents, biological control, innate plant immunity, plant growth promotion

## Abstract

**Simple Summary:**

This study evaluated two *Bacillus halotolerans*-beneficial strains (Cal.l.30 and Cal.f.4) for the levels of their compatibility and their potential to be effective biofertilizers and in parallel biocontrol agents. We investigated their plant growth-promoting effect and ability to induce the expression of defense-related genes in the leaves of young tomato seedling plants, applying them individually or in combination under in vitro and greenhouse conditions using seed biopriming and soil drenching as inoculum delivery systems. The results showed that *B. halotolerans* strains (singly or in combination) are promising bioproducts for sustainable horticulture as they combine both direct antifungal activity against plant pathogens and the ability to prime plant immunity and enhance plant growth.

**Abstract:**

Evaluating microbial-based alternatives to conventional fungicides and biofertilizers enables us to gain a deeper understanding of the biocontrol and plant growth-promoting activities. Two genetically distinct *Bacillus halotolerans* strains (Cal.l.30, Cal.f.4) were evaluated for the levels of their compatibility. They were applied individually or in combination under in vitro and greenhouse conditions, using seed bio-priming and soil drenching as inoculum delivery systems, for their plant growth-promoting effect. Our data indicate that application of Cal.l.30 and Cal.f.4 as single strains and as a mixture significantly enhanced growth parameters of Arabidopsis and tomato plants. We investigated whether seed and an additional soil treatment with these strains could induce the expression of defense-related genes in leaves of young tomato seedling plants. These treatments mediated a long lasting, bacterial-mediated, systemic-induced resistance as evidenced by the high levels of expression of *RP3*, *ACO1* and *ERF1* genes in the leaves of young tomato seedlings. Furthermore, we presented data showing that seed and soil treatment with *B. halotolerans* strains resulted in an effective inhibition of *Botrytis cinerea* attack and development on tomato leaves. Our findings highlighted the potential of *B. halotolerans* strains as they combine both direct antifungal activity against plant pathogens and the ability to prime plant innate immunity and enhance plant growth.

## 1. Introduction

To meet the growing demand for humans’ food and animals’ feed, increasing agronomic inputs such as chemical fertilizers and pesticides raise public concerns on the environment and human health. Hence, new alternatives have been sought that would lead to an increase in agricultural production via eco-friendly and sustainable strategies [1]. Thus, the use of natural compounds and plant-beneficial microorganisms have been proposed [1]. These beneficial microorganisms may be bacteria associated with below- or above-ground plant areas, or bacteria inhabiting the plant endosphere (endophytic bacteria) and/or the rhizosphere (rhizobacteria), collectively known as plant growth-promoting bacteria (PGPB) [2]. These PGPB are equipped with a number of traits through which they improve plant growth in diverse agricultural settings. The mechanisms of plant growth promotion include the facilitation of nutrient acquisition and the synthesis of plant hormones, restriction of pathogens’ growth or even the elimination of pathogens through antibiotic production and/or enzyme production, competition for nutrients and appropriate colonization of niches at the root surface [1]. Furthermore, PGPB protect plants against pathogens by activating physiological, biochemical and molecular defense responses in the plant [3].

Inoculation of plants with beneficial microorganisms can be performed with a single isolate or with more than one, called co-inoculation. In co-inoculation, beneficial microorganisms may interact synergistically, increasing the efficiency of exerted plant growth-promoting activities [4,5]. For the inoculation of plants with beneficial microorganisms, different methods are being employed including seed treatment, root dipping, soil drenching and foliar spraying inoculation, with seed inoculation or soil drenching being the most widely used methods in various agricultural settings [6]. The colonization of host plant tissues (e.g., roots) by PGPB can prime, sensitize or activate plants’ innate immune defense system and, thus, can render the plant capable of responding quickly and strongly as soon as the pathogen is perceived [7,8]. In general, the priming process sensitizes or activates the expression of the genes of different hormonal pathways [9,10]. Two types of induced defenses were proposed, ISR (induced systemic resistance) and SAR (systemic acquired resistance), depending on the hormone implicated and the type of elicitor [10]. Accordingly, ISR is initiated by beneficial rhizobacteria or other non-pathogenic microorganisms, while SAR is induced by pathogens or chemical compounds [10]. The signal transduction pathway of ISR is regulated by the Jasmonic acid/Ethylane (JA/ET) pathway and is associated with the expression of JA/ET marker genes such as ACC oxidase (ACO), which is involved in the final step for ethylene biosynthesis (*ACO1*) and genes coding for ethylene response factor 1 (*ERF1*), a transcription factor involved in the regulation of gene expression responding to biotic and abiotic stresses [10]. SAR, on the other hand, is controlled by the SA-dependent signaling pathway and characterized by the expression of SA marker genes encoding pathogenesis-related (PR) proteins [10]. ISR is commonly regarded as SA-independent and develops without the accumulation of PR proteins [7]; however, there are a few exceptions where there is a positive SA-JA interplay, resulting in the simultaneous induction of SA and JA signaling pathways. For example, the colonization of *Arabidopsis thaliana* by *Enterobacter radicincitans* DSM 16,656 enhanced plant growth and induced the priming of plant defense responses, as was evidenced by the accumulation of ISR or SAR marker genes, *PR1*, *PR2*, *PR5* and *PDF1.2* as compared to untreated plants [11]. The presence of *Bacillus amyloliquefaciens* MBI600 in the in-tomato rhizosphere or phyllosphere activated the induction and interplay of ISR and SAR, as was evidenced by the up-regulation of salicylic acid, jasmonic acid or ethylene-responsive marker genes [12].

The most promising PGPB that are able to function as priming agents and plant growth-promoting agents belong to *Bacillus* and *Pseudomonas* species [5,13,14]. In most cases, the inoculation of plants with beneficial microorganisms has been performed with a single isolate. However, recent studies have demonstrated that inoculation with mixtures of more than one isolate may interact synergistically, resulting in improved plant development and enhanced priming effect compared to single strain inoculations [5,15].

The aim of this study was to combine the novel endophytic *B. halotolerans* strains Cal.l.30 and Cal.f.4, two promising biological control agents, and evaluate their plant growth-promoting and plant protection abilities as mono- and co-cultures. Cal.l.30 and Cal.f.4 were selected ahead of other isolates due to their ability to combine direct plant growth effect and strong antimicrobial activity against several plant pathogens with the secretion of novel secondary metabolites [16,17]. Strains were tested for their compatibility under in vitro conditions and their co-colonization capacity on *A. thaliana* roots. We used seed bio-priming and soil-drenching methods as inoculum delivery systems to apply both strains individually or in combination on tomato plants and evaluate their plant growth promotion. We further studied the activation of tomato plant defense mechanisms after the application of inoculants through seed priming and soil drenching and evaluated their ability to protect plants against gray mold disease. Finally, we evaluated the direct suppression of *B. cinerea* under ex vivo application on detached grape berries. 

## 2. Materials and Methods

### 2.1. Bacterial Preparation

Both *B. halotolerans* Cal.l.30 and *B. halotolerans* Cal.f.4 are natural endophytes isolated from *Calendula officinalis*, as recently described by Tsalgatidou et al., 2023 [16]. Routinely, both strains were inoculated in 5 mL liquid nutrient broth (NB) medium and incubated overnight with shaking on an orbital platform (200 rpm) at 30 °C. Bacteria were maintained on solid nutrient agar (NA) plates (1.5% *w/v* agar) until further use. 

### 2.2. Compatibility Assays In Vitro

The in vitro compatibility of Cal.l.30 and Cal.f.4 was studied by the modified mixed colony spot [18] and pairwise interaction methods [19]. Cal.l.30 and Cal.f.4 have different colonial morphology and could be differentiated based on this. To further support the results, the rifampicin-resistant strain of Cal.l.30 (Cal.l.30-rif) was used to evaluate coexistence of Cal.l.30 and Cal.f.4. A mixed colony spot was performed between bacterial strains Cal.l.30-rif and Cal.f.4 onto an NA medium in 5 cm diameter Petri dishes. Both bacterial strains, each containing ~10^8^ CFU/mL (OD_600_ = 0.5)—based on standard curves of each bacterial strain—were equally combined, creating a 1:1 mixture. After mixing thoroughly, an aliquot of 10 μL of the mixture (5 μL of each bacterial strain) was spot inoculated on the agar surface and incubated at 30 °C for five days. Colony growth was captured and measured on a daily basis until the fifth day of observation. The bacterial cell density of mono- and co-culture colonies was calculated daily after pouring 1 mL of sterilized ddH_2_O onto petri plates, excluding the cells from the NA medium with a Pasteur pipette and, finally, serially diluting and plating 100 μL of the resuspended cells on NA plates supplemented with or without rifampicin. Plates were incubated overnight at 30 °C. Experiments were conducted in triplicate and repeated three times. 

The pairwise interaction method was performed between *B. halotolerans* Cal.l.30 and Cal.f.4 strains onto the NA medium in 5 cm Petri dishes. An aliquot of 5 μL of each bacterial strain of cell suspension of 10^8^ CFU/mL (OD_600_ = 0.5) was spot inoculated at a 1 cm distance between them. Each strain was singly inoculated as a control treatment. Plates were incubated at 30 °C for 7 days and images were taken on a daily basis. For each treatment, four biological replicates were run in three technical replicates.

### 2.3. Bacterial Interactions on A. thaliana Plantlets

Bacterial strains Cal.l.30 and Cal.f.4 were studied for their in planta compatibility by the root colonization of *A. thaliana* seedlings and the plant growth effect, as mono- and co-cultures. *A. thaliana* seeds were surface sterilized and grown on solid half-strength Murashige and Skoog (½ MS) medium (Duchefa Biochemie, Haarlem, The Netherlands) supplemented with 0.5% (*w*/*v*) sucrose and 0.8% (*w*/*v*) bacteriological agar (Sigma, Burlington, MA, USA) as previously described by Tsalgatidou et al., 2023 [16], until further use. 

To evaluate the simultaneous *A. thaliana* root colonization ability of Cal.l.30 and Cal.f.4 as a mixture, a modified protocol of Beauregard et al., 2013 [20] was conducted. Eight-day-old seedlings grown on solid ½ MS medium were transferred to Eppendorf’s containing 300 mL ½ MS medium, inoculated with bacterial cells of ~10^8^ CFU/root (OD_600_ = 0.5) of mono- (Cal.l.30, Cal.f.4) and co-cultures (Cal.l.30 + Cal.f.4) and put on an orbital shaker at 100 rpm for 18 h. For this experiment, the rifampicin resistant strain Cal.l.30-rif was used. To determine colonization ability, plants were removed from each tube, washed two times in 1 mL of ½ MS after shaking thoroughly for 30 sec and finally washed in 1 mL of ½ MS with vortex agitation for 1 min to remove all cells slightly attached to the roots. Cell-containing solution was serially diluted with ½ MS before plating on NA medium supplemented with rifampicin, only for Cal.l.30-rif determination. The CFU/root was determined 24 h after incubation at 30 °C. For each treatment, six biological replicates were run in three technical replicates. Single strain bacterial cells’ colonization was visualized under an Olympus BX40 microscope.

The growth effect on *A. thaliana* morphological characteristics was determined after bacterial mono- (Cal.l.30 and Cal.f.4) and co-culture (Cal.l.30 + Cal.f.4) inoculation at 3 cm distance from root tips, as previously described by Tsalgatidou et al., 2023 [16]. Both bacterial strains, each containing ~10^8^ CFU/mL (OD_600_ = 0.5), were equally combined, creating a 1:1 mixture, and an aliquot of 10 μL cell suspension was inoculated. For each treatment six biological replicates were run in three technical replicates. Seedlings’ fresh weight, primary root length, lateral roots and root hairs were determined as previously described by Tsalgatidou et al., 2023 [16]. 

### 2.4. Pot Experiment Design

#### 2.4.1. Tomato Seed Bio-Priming and Soil Drenching with Cal.l.30 and Cal.f.4 as Single- and Dual-Strain Inoculant

Tomato seeds (*Solanum lycopersicum* var. Chondrokatsari Messinias) were bacterized with *B. halotolerans* strains Cal.l.30 and Cal.f.4 as a single-strain formulation and as a mixture under greenhouse conditions with a completely randomized block design with three replicates. Tomato seeds were kindly provided by Dr Costas Delis (Department of Agriculture at the University of the Peloponnese). Sterilization and bio-priming of tomato seeds were conducted, as previously described, by Thomloudi et al., 2021 [21]. Briefly, tomato seeds were surface sterilized by immersion in a 3% (*v*/*v*) aqueous solution of commercial bleach (5% sodium hypochlorite solution) for 6 min, rinsed six times in double distilled water and air-dried in a UV-sterilized laminar flow cabinet. Dried seeds were immersed in a solution containing a single strain and a 1:1 mixture of bacterial suspension adjusted to ~10^8^ CFU/mL (OD_600_ = 0.5), supplemented with 1% (*w*/*v*) Carboxyl Methyl Cellulose (CMC) (Sigma-Aldrich^®^, Merk KGaA, Burlington, MA, USA). Control seeds were immersed in a solution containing only CMC. Seeds were incubated for 1.5 h at 25 °C under agitation and then air-dried in a UV-sterilized laminar flow cabinet, until sowing in 7 × 7 cm^2^ pots (one plant/pot) using a substrate of peat and perlite (2:1 *v*/*v*). 

Additional bacterial application was done by the soil drenching (10 mL/pot) of Cal.l.30, Cal.f.4 and Cal.l.30 + Cal.f.4 (1:1 mixture) suspension adjusted to ~10^8^ CFU/mL, at 15 d 30 d after sowing. The pot experiment was conducted in a greenhouse (average air temperature 25 °C, relative air humidity 60% and an average 10½ h photoperiod) from October to December of 2019. Plant growth parameters (shoot fresh and dry weight, root fresh and dry weight and shoot length) were recorded 45 days after sowing (das). Plants were captured with a digital camera and shoot length was measured using ImageJ software v.1.8.0 [22].

#### 2.4.2. Rhizosphere Colonization Assay

Bacterial rhizosphere colonization and soil survival were estimated at two time points: at 15 das, to measure the number of cells of each treatment in the rhizosphere soil with bio-priming being the only inoculation system, and at 45 days after two extra bacterial inoculations were conducted through soil drenching. For this experiment, the rifampicin resistant strain Cal.l.30-rif was used. Plants were carefully uprooted and, after removing most of the soil from the roots, one gram of the remaining soil tightly adhered to the roots was taken. The soil was transferred to a falcon tube containing 10 mL of sterilized ddH_2_O and vortexed for 1 min. The bacterial population of each treatment was enumerated at each sample day by a standard dilution plate count method on NA medium. The colonies of Cal.l.30-rif were counted by supplementing the NA medium with rifampicin. The CFU/g of soil were determined 24 h after incubation at 30 °C. For each treatment, four biological replicates were run in three technical replicates. 

#### 2.4.3. *B. cinerea* Infection Assay

After 45 days of bacterial inoculation through seed bio-priming and additional soil drenching applications at 15 and 30 das, *B. cinerea* was inoculated on tomato leaves in vivo. A droplet of 10 μL of spore suspension (5 × 10^5^ CFU/mL) was inoculated on one side of the midvein of the leaves (one leaf/tomato plant). Inoculated plants were covered with transparent plastic to maintain high humidity and plants were left in the greenhouse for five more days. After 5 days post infection (dpi), the lesion area created by *B. cinerea* on tomato leaves was measured and disease severity index (DSI %) was calculated as described by Lee et al., 2006 [23]. The lesion area was measured using the ImageJ software v.1.8.0 analysis tool [22]. Disease severity (DS %) was calculated as the percentage of infected leaf area on which a 0-to-4 rating scale was created, with 0 = healthy leaves, 1 = 0.1–5%, 2 = 5.1–20%, 3 = 20.1–40% and 4 = 40.1–100% lesion area [23]. The disease severity index (DSI %) was estimated using the above severity scale and based on the following formula: % Disease Severity Index (DSI) = [∑(number of infected leaves in a specific value of disease rating scale × corresponding value of disease scale)/(maximum rating scale value × total number of leaves rated)] × 100.

### 2.5. Quantitative Analysis of Defense-Related Gene Expression

The induction of defense-related genes after bacterial formulation (Cal.l.30, Cal.f.4 and Cal.l.30 + Cal.f.4) through seed bio-priming and both seed bio-priming and an additional root drenching (15 d), was evaluated by quantitative real-time PCR. Tomato plantlets’ leaf samples were collected for RNA extraction 15 das of bio-primed seeds and 24 h after an additional bacterial soil drenching application. 

Total RNA was extracted from tomato leaves (100 μg/sample) of four different treatments (non-inoculated plants (control), inoculation with *B. halotolerans* Cal.l.30, inoculation with *B. halotolerans* Cal.f.4 and inoculation with the mixture of *B. halotolerans* strains Cal.l.30 + Cal.f.4) at two different time points (15 das of bacterial bio-primed seeds and 24 h after additional soil drenching) following the protocol from the manufacturer, Macherey-Nagel (NucleoSpin^®^ RNA Plus, Macherey-Nagel, Loughborough, Leicestershire, UK). DNA was DNAase eliminated and cDNA was synthesized with 3 μg of purified RNA using the PrimeScript 1st strand cDNA, Synthesis Kit (Takara Bio, Kusatsu, Shiga, Japan), according to the manufacturer’s instructions. The conditions of RT-qPCR and the relative quantification of specific mRNA levels were performed using PowerUp SYBR Green Master Mix Assay (Applied Biosystems, Waltham, MA, USA)) containing 1 μL cDNA template, 10 μL SYBR GREEN PowerUp mix and 2 μL each of 10 μM tomato gene-specific primer pair (Appendix A). The cycling conditions were as follows: pre-incubation at 95 °C for 2 min, followed by 40 cycles at 95 °C for 15 sec, at 60 °C for 15 sec and at 72 °C for 1 min. *Ubi3* gene coding for ubiquitin [24] was used as a housekeeping gene and for the normalization of gene expression. Primers used against genes coding for pathogenesis-related proteins PR2; *PR2* [25] and PR3; *PR3* (Beacon Designer v.7.9), for proteins related to ethylene synthesis and response such as ERF1; *ERF1* [26] and ACO1; *ACO1* (Beacon Designer v.7.9), and the lipoxygenase A encoding gene *TomloxA* [27] that catalyzes the first step in the octadecanoid pathway as a response to biotic and abiotic stress, were selected. 

The efficiency of the reaction for each target gene derived from LinRegPCR software [28] through linear analysis regression on the logarithm of the fluorescence intensity values in each cycle reaction. Relative levels of target gene transcripts for each sample were calculated based on the formula derived from the Pfaffl equation: Eref^Cref^/Ectarget^cttarget^ [29]. 

### 2.6. Detached Fruit Assay

The direct biological control ability of *B. halotolerans* Cal.l.30 and *B. halotolerans* Cal.f.4 as individual strains and as a consortium was evaluated on detached grape berries (*Vitis vinifera* L. cv. Sultanina) against gray mold disease caused by *B. cinerea* as previously described by Tsalgatidou et al., 2022 [17]. Briefly, an aliquot of 10 μL of single strain (Cal.l.30 and Cal.f.4) and a 1:1 mixture (Cal.l.30 + Cal.f.4) of bacterial suspension adjusted to ~10^8^ CFU/mL (OD_600_ = 0.5) were inoculated on a 3 mm wound on the surface-sterilized grape berries (20 fruit/replicate). The grapes were incubated for 1 h at room temperature until infection with 10 μL of *B. cinerea* spore suspension (~1 × 10^5^ CFU/mL). The grapes were placed into plastic boxes and transferred to a dark growth chamber at 25 °C for 5 d. After 5 dpi, the lesion area created by *B. cinerea* was measured and the disease severity index and disease incidence were calculated as previously described by Tsalgatidou et al., 2022 [17]. The lesion area was measured using ImageJ software v.1.8.0. Finally, the colonization ability on grapes of both single strains (Cal.f.4 and Cal.l.30) and their mixture was evaluated at 24 h post inoculation as previously described by Tsalgatidou et al., 2022 [17]. For this experiment, the rifampicin-resistant strain Cal.l.30-rif was used to determine colonization on grapes.

### 2.7. Statistical Analysis

Statistical analysis was performed using IBM SPSS Statistics for Windows, version 25 (IBM Corp., Armonk, NY, USA). Plots were created with Sigma Plot, version 12.0 (Systat Software, San Jose, CA, USA). In bacterial compatibility assays, *A. thaliana* and tomato pot experiments were performed and in detached grapes assays, statistical analysis was performed with ANOVA followed by Tukey’s honestly significant difference (HSD) test (*p* < 0.05) to allow for comparisons among all means. To evaluate the bacterial cell number, data were transformed to the logarithmic scale followed by Tukey analysis. Plots indicate mean values with standard deviation as error bars. Different letters represent statistically significant differences among treatments. Data expressed as percentages were arcsine transformed prior to Tukey analysis (*p* < 0.05). For gene expression analysis, statistical analysis was performed on the logarithmic values of the relative expression (on ΔCt) to ensure the assumptions of parametric statistical analysis where data follow a normal distribution and homogeneity of variances. Graphs indicate the mean of gene expression for each treatment and a 95% confidence interval (Mean ± CI). 

## 3. Results

### 3.1. Co-Existence of B. halotolerans Cal.l.30 and Cal.f.4 in In Vitro Compatibility Assays

Members of a bacterial consortium are considered as compatible when they do not inhibit each other’s growth during their in vitro co-culture and/or in rhizosphere colonization competition assays [15]. In this context, the in vitro compatibility of the strain mixture involving Cal.l.30 and Cal.f.4 was evaluated by determining the level of co-existence during growth in colony and biofilm modes in a series of co-culture experiments. First of all, we dropped, inoculated, each strain either singly or as bacterial mixture (1:1) on an NA agar plate and measured the colony morphology (Figure 1A) as well as the number of bacterial cells (Figure 1D) in each of the evolving colonies. The results shown in Figure 1B,D indicate that the number of Cal.l.30 and Cal.f.4 cells in the spot-inoculated two-strain mixture-developing colony remained almost identical when compared to the number of cells found in each monoculture, suggesting that Cal.l.30 and Cal.f.4 strains grown in close proximity do not inhibit the growth of each other and, thus, can co-exist, occupying the same niche under the given environmental conditions.

We also carried out experiments in which two different colonies (Cal.l.30 and Cal.f.4) were inoculated on NA agar plates with a separation of 10 mm, and their external (in the direction opposite to the other colony) and internal (in the direction of the other colony) radii were periodically depicted and measured for 7 days. As shown in Figure 2A, when Cal.l.30 and Cal.f.4 strains are inoculated in close proximity to each other, although there is a slight decrease of the Cal.f.4 internal radius as compared to the external one, the area covered by the interacting macrocolonies was largely unaffected as compared to singly grown macrocolonies (Figure 2B,C). These data suggested that both strains interacted, and no antagonistic relationship was found, suggesting that Cal.f.4 coexisted well with Cal.30.

Compatibility of the two *B. halotolerans* strains was also evaluated by investigating whether antagonism between two *B. halotolerans* strains occurring on the root may affect population density of each strain, using an in vitro root colonization assay. As shown in Figure 3B,C, both bacterial strains can successfully colonize the root surface of *A. thaliana* plantlets either as single cells or as aggregates (black arrows). Inoculation of Cal.l.30 and Cal.f.4 as single cultures on hydroponic grown *A. thaliana* Col-0 resulted in similar levels (3–4 × 10^4^ CFU/root) of colonization by each species, whereas inoculation with a mixture (1:1) of both strains resulted in increased and cumulative colonization of roots by both strains (6–7 × 10^4^ CFU/root from each strain) (Figure 3A). These data suggested that both bacterial strains could co-exist on roots in a synergistic manner. 

### 3.2. Application of B. halotolerans Strains in Combination and Individually Enhanced the Growth Parameters of Arabidopsis Plants Grown under In Vitro Conditions

As a first step towards evaluating the plant growth effect of *B. halotolerans* strains, we evaluated the capacity of Cal.l.30 + Cal.f.4 consortium to promote plant growth in comparison with single-strain applications, using an in vitro *A. thaliana* co-cultivation approach (Figure 4A). Treatment with Cal.l.30, Cal.f.4 and a Cal.l.30 + Cal.f.4 mixture significantly increased the growth of *A. thaliana* as evidenced by the enhanced values of all growth parameters measured (fresh weight, number of lateral roots and root hairs) (Figure 4B). Treatment of Arabidopsis by the two-strain mixture resulted in plants with higher total fresh weight and lateral roots compared to the single-strain formulations and the control plantlets (Figure 4B). The single-strain formulation of Cal.l.30 resulted in the highest root hair number compared to the other two formulations.

### 3.3. Application of B. halotolerans Strains as a Mixture and Individually Enhanced Growth Parameters of Tomato Plants under Greenhouse Conditions

The plant growth promotion capacity of Cal.l.30 and Cal.f.4 was investigated when single strains and bacterial mixtures were applied as tomato seed treatment followed by two additional soil drenching treatments at 15 and 30 das under greenhouse conditions. Plants were harvested at 45 das and growth parameters were measured (total shoot and root, fresh and dry weight and shoot length). As illustrated in Figure 5, seed and soil treatment with Cal.l.30 and Cal.f.4, individually or in combination, resulted in a pronounced growth of tomato plants’ growth parameters compared to untreated plants (control). The results presented in Figure 5 revealed that compared to the control, maximum shoot and root biomass was recorded in plants treated with Cal.l.30, followed by plants treated with Cal.l.30 + Cal.f.4 (Figure 5C,D). Examination of plants’ rhizospheric bacterial populations revealed that Cal.l.30 and Cal.f.4 remained viable in interactions with roots at 15 and 45 das, supporting their long lasting effects to plants.

### 3.4. Seed Priming and Additional Soil Drenching with Cal.l.30 and Cal.f.4 Strains Singly or in Combination Protect Tomato Plants against Gray Mold Disease

To evaluate whether seed and soil treatment with Cal.l.30, Cal.f.4 and their combination (Cal.l.30 + Cal.f.4) induce systemic resistance of tomato against gray mold disease, disease symptoms on leaves caused by *B. cinerea* were examined at 5 days after infection with the pathogen (Figure 6B). Five days after pathogenic fungus inoculation, leaves belonging to Cal.l.30, Cal.f.4 and Cal.l.30 + Cal.f.4-treated plants presented minor disease symptoms (brown spots and necrotic lesion) around inoculating loci, compared to untreated plants (Figure 6D). The lesion area was 1.8 cm^2^, 2.1 cm^2^ and 1.6 cm^2^ in Cal.l.30, Cal.f.4 or Cal.l.30 + Cal.f.4-treated plants, respectively, compared to 8.2 cm^2^ in untreated plants, suggesting that Cal.l.30 + Cal.f.4 treated plants exhibited the best inhibition disease development (Figure 6C).

To further identify the biocontrol efficacy of Cal.l.30, Cal.f.4 and Cal.l.30 + Cal.f.4-treated plants against *B. cinerea*, the disease severity index was calculated. As presented in Figure 6A, the disease severity index (%) of control plants is 78.0%, compared to the low values of Cal.l.30, Cal.f.4 and Cal.l.30 + Cal.f.4-treated plants ranging between 25–26%, suggesting that the bacterial mixture and the single strains were equally effective in controlling gray mold disease.

### 3.5. Activation of Long-Lasting Defense-Related Genes of Tomato Plantlets

Besides the impact of *B. halotolerans* strains in control efficacy against *B. cinerea*, to understand whether seed bio-priming and soil drenching with Cal.l.30, Cal.f.4 and Cal.l.30 + Cal.f.4 could induce the expression of the long-lasting defense-related genes, RT-qPCR analysis was conducted. Therefore, we analyzed the expression of the SA-responsive genes *PR3* (encoding the PR protein 2 β-1,3-glucanase) and *PR2* (encoding the PR protein 2 β-1,3-glucanase), the JA synthesis-related gene TomLoxA (encoding for lipoxygenase A), the ET-responsive gene ERF1 and the ethylene biosynthesis-related gene ACO1 in the leaves of young plantlets after being seed bio-primed with Cal.l.30, Cal.f.4 and Cal.l.30 + Cal.f.4 (15 das) and after additional application through root drenching with Cal.l.30, Cal.f.4 or Cal.l.30 + Cal.f.4 (24 h post additional soil drenching). 

As illustrated in Figure 7A, the expression level of *ERF1*, *ACO1* and *PR3* at 15 das were strong induced in plants inoculated with Cal.l.30 through seed bio-priming in comparison to the control (unprimed seeds). Specifically, gene transcript levels of *ERF1*, *ACO1* and *PR3* were up-regulated 1.98-, 1.68- and 4.12-fold, respectively, compared to the control (Figure 7C). Tomato plants raised from Cal.f.4 bio-primed seeds presented up-regulated gene transcript levels of *ACO1* and PR3 genes (1.63- and 3.49-fold, respectively), while *ERF1* expression was down-regulated (Figure 7C). Bio-primed plants with a Cal.l.30 + Cal.f.4 consortium presented a significant up-regulation (2.58-fold) of *PR3* transcripts compared to control plants, while *ERF1* and *ACO1* gene expressions presented a similar level in gene expression in both non- and bio-primed plants (Figure 7C). Expression of *PR2* was only observed in plants inoculated with Cal.l.30 + Cal.f.4 consortium (Figure 7A). These results indicated that defense mechanisms were less activated in plants with combined bacterial priming treatment compared to the individual treatments. 

The gene expression levels of *ERF1*, *ACO1*, *PR2*, *PR3* and *TomLoxA* were significantly differentiated after both seed bio-priming and additional bacterial formulation via soil drenching at 15 das (Figure 7B). As reported in Figure 7D, transcript levels of *ERF1*, *ACO1* and *PR3* genes of all bacterial inoculated treatments were up regulated compared to untreated plants. Specifically, in Cal.l.30-treated plants, transcript levels of the ERF1, ACO1 and *PR3* genes were up-regulated 2-, 1.8- and 3-fold when compared to control plants, while in Cal.f.4-inoculated plants transcript levels were up-regulated 2-, 1.2- and 4.4-fold, respectively. Finally, the expression levels of the *ERF1*, *ACO1* and *PR3* genes after consortium’s treatment were up-regulated 1.7-, 1.6- and 6-fold (Figure 7D). Expression of the *TomLoxA* gene was only recorded in plants inoculated with Cal.l.30, while *PR2* gene expression was only triggered after inoculated plants with Cal.l.30 and the consortium (Cal.l.30 + Cal.f.4) (Figure 7B). These results indicated that soil drenching with *B. halotolerans* strains singly or as a two-strain mixture efficiently activated the expression of JA/ET- and SA-responsive genes in the leaves of tomato plants under greenhouse conditions.

### 3.6. B. halotolerans Strains Cal.l.30 and Cal.f.4 (Singly or as a Mixture) Suppress Grape’s Postharvest Gray Mold Disease

Although the efficacy of *B. halotolerans* Cal.l.30 and *B. halotolerans* Cal.f.4 as single-strain biological control agents against *B. cinerea* in ex vivo applications has already been tested in previous studies [16,17], the application of these strains as a consortium is studied here for the first time. All three applications were tested on detached grape berries in a small-scale storage experiment against *B. cinerea*. Both single bacterial applications, as well as their mixture, successfully colonized grapes at the point of application, showing a similar cell number (1–3 × 10^7^ CFU/wound).

As presented in Figure 8A,B, Cal.l.30 and Cal.f.4 single-strain applications, as well as their 1:1 mixture, presented a strong inhibition against *B. cinerea* by reducing both the number of infected grapes and the brown lesion area. Grapes infected only with the pathogenic fungus developed extended brown lesion and visible mycelia growth around the point of formulation and reached 100% infection rate for *B. cinerea* (Figure 8A). Negative control grape berries that were wounded artificially but not infected with the fungal pathogen appeared to be unaffected after 5 days of storage. Both single-strain formulations significantly reduced the disease severity index, with strain Cal.l.30 being the most efficient with 31% of DSI and 81% of DI, in comparison to the 46% DSI and 97% DI of Cal.f.4 (Figure 8C,D). However, treatment with Cal.l.30 + Cal.f.4 synergistically suppressed disease severity, reaching the lowest DI value of 72% and DSI value of 28% at 5 day of incubation, presenting, along with the Cal.l.30 single-strain formulation, the most impressive results.

## 4. Discussion

Plant growth-promoting bacteria (PGPB) can provide multiple benefits in agriculture by mediating enhancement in crop productivity and nutrient content and suppressing the growth of pathogens. Although members of the *B. subtilis* species complex have been extensively studied for their ability to promote vegetative plant growth, reduce disease incidence and promote plant tolerance against fungal and bacterial pathogens by inducing the systemic resistance of host plants, the quest for new Bacillus-based biological control agents (BCAs) along with effective methods for their delivery is an ongoing process [30,31].

*B. halotolerans* strains Cal.l.30 and Cal.f.4 have a wealth of plant growth-promoting traits and exhibit a strong antimicrobial activity against fungal pathogens through the production of multiple antifungal metabolites [16,17]. The present study provides evidence that Cal.l.30 and Cal.f.4 could form a compatible consortium under in vitro, ex vivo and greenhouse conditions. The efficient coexistence of Cal.l.30 and Cal.f.4 was observed when both strains successfully colonized *A. thaliana* roots in in vitro application, presenting a greater adhesion of their cells on the roots when applied as a mixture than as single strains. Cal.l.30 + Cal.f.4 mixture synergistically promoted *A. thaliana* plant growth parameters under in vitro conditions, compared to single treatments. 

Recent studies have demonstrated that seed bio-priming and soil drenching are the most promising methods for delivering the beneficial microbes to the host plant when PGPB are evaluated for their plant growth-promoting activity under greenhouse or field conditions [32,33]. Furthermore, it has been demonstrated that tomato, cow pea and mung bean seed bio-priming followed soil drenching induced significant plant growth parameters as compared to single treatment [34,35]. In this regard, in our studies, we evaluated the plant growth effect of *B. halotolerans* Cal.l.30 and *B. halotolerans* Cal.f.4 when applied to tomato, singly or in combination, through seed bio-priming and additional soil drenching. Our data demonstrated that tomato seed bio-priming followed by soil drenching with all inoculum formulations (Cal.l.30, Cal.f.4 and Cal.l.30 + Cal.f.4) gave better results in plant growth-promotion parameters compared to untreated plants. These findings are in line with previous studies showing that tomato plants inoculated through seed treatment and soil drenching methods with *Pseudomonas* sp. strain S3 led to more improved morphological characteristics (root/shoot length, fresh weight and dry weight) compared to a single-method treatment [36]. It should be pointed out that, in contrast to Arabidopsis studies where the bacterial mixture showed a better performance compared to the individual strains, when applied on tomato plants the effect of the Cal.l.30 + Cal.f.4 mixture was more pronounced when compared to the single-strain Cal.f.4 treatment, and somewhat inferior compared to treatment with Cal.l.30, suggesting that consortia may provide better plant growth-promoting effects than some individual strains. This is in line with another study, where it was observed that the effect of seed bio-priming and soil drenching in biomass-promoting effects on *Withania somnifera* is more pronounced when *B. amyloliquefaciens* and *Pseudomonas fluorescens* were applied as a mixture compared to individual treatments [5]. 

These findings highlights that the plant growth-promoting effect observed in tomato plants treated with Cal.l.30 and Cal.f.4 could partially be due to the known plant growth-promoting traits of Cal.l.30 and Cal.f.4, such as production of IAA and siderophores, phosphate solubilization [16] and/or due to the PGPB-mediated long-lasting induction of the expression of genes related to plant growth and development [37,38,39].

Our data indicated that tomato plants treated with Cal.l.30 and Cal.f.4 through seed bio-priming and soil drenching showed a defense-priming effect; plants raised from Cal.l.30 and Cal.f.4-treated seeds showed elevated levels of transcripts of SA and JA/ET defense-related genes, while plants raised from the Cal.l.30 + Cal.f.4 mixture-treated seeds showed enhanced levels of SA-responsive genes. However, plants raised from Cal.l.30, Cal.f.4 or Cal.l.30 + Cal.f.4 bio-primed seeds that also received a soil drenching application displayed a relatively high level of expression of SA and JA/ET defense-related marker genes, indicating that Cal.l.30 and Cal.f.4 strains, singly or in combination, do not only sensitize the plants’ SA and JA/ET immune responsive systems but also enable the plant to enter the defense-priming state. In this regard, plant’s priming is able to maintain a long-lasting state of readiness that does not confer resistance per se, as the accumulation of defense-related transcripts is below the threshold for effective resistance, but rather allows accelerated activation of inducible defense as soon as a pathogen or pest invasion occurs [40,41]. 

These findings are consistent with previous studies where it was reported that seed treatment followed by soil drenching with certain beneficial ISR-inducing microorganisms, including *Bacillus* sp. and *Trichoderma* spp., reprograms their transcriptional apparatus and up-regulates inducible defenses (enhanced expression of SA- and/or ET-defense genes and/or genes coding antioxidative enzymes and enzymes involved in callose deposition and lignification) [9,37,38,42,43,44,45,46].

The duration of the priming state in tomato plants raised from bio-primed seeds or root-drenched plants is not known. Priming-mediated defenses can be durable and maintain throughout the plant’s life cycle and can even be transmitted to the subsequent generation [9]. However, recent studies demonstrated that the inoculum application mode, and most importantly the number of applications, is important for the duration and efficacy of PGPM (plant growth-promoting microorganism) priming-mediated defense [47]. Seed dressing and soil drenching with species of *B. amyloliquefaciens* FS6 [48], *Bacillus subtilis* JW-1 [49], *Bacillus thuringiensis* SE and *Pseudomonas aeruginosa* KA19 [50] and *Bacillus velezensis* SZAD2 [51], were found to be effective methods for the control of fungal pathogens. However, when seed treatment was followed by soil drenching, the biocontrol efficacy appeared to be more effective than either treatment alone [48,49]. Suppression of *Rhizoctonia solani* was found more effective after two soil drenching applications of *B. amyloliquefaciens* FZB42 at short time intervals, compared to single treatment of tomato plants through soil drenching [52]. Similarly, suppression of potato virus Y was found more in tomato plants subjected to *B. amyloliquefaciens* MBI600 triple soil drench application compared to a single application through soil drenching prior to germination [53]. In addition, suppression of *Fusarium* was found more when tomato plants were treated with MBI600 dual drenching (soil drenching prior to germination followed by additional soil drenching), compared to single soil drenching just after sowing [33,46]. Taken together, these data suggested that the back-to-back application of ISR-inducing PGPB (e.g., seed bio-priming followed by soil drenching, seedling treatment), as compared to a single ISR-inducing PGPB application, may prolong the priming defense state of the host plant and, thus, keep the elevated levels of gene expression of defense-related genes. This offers better prospects for an accelerated activation of inducible defenses as soon as an invasion occurs [39,40].

In our studies, the biocontrol efficacy of Cal.l.30, Cal.f.4 and Cal.l.30 + Cal.f.4 was evaluated on tomato plants raised from bio-primed seeds that received two additional doses of bacteria by soil drenching and then challenged with *B. cinerea*. Under this scheme of inoculum application and seed bio-priming followed by two soil drenchings at short time intervals, tomato plants may sustain the primed defense state (elevated levels of gene expression of SA and ET/JA defense-related genes), thus possessing the means to develop a fast and effective defense response upon facing the ingressive *B. cinerea*. Based on their disease-suppression abilities, when plants were treated with Cal.l.30 + Cal.f.4, a significant reduction in the plant disease severity was achieved with respect to the control, but this result was not significantly different than those observed in the individual-strain treatments. However, plants treated with Cal.l.30 + Cal.f.4 exhibited better disease inhibition development around the infection zone compared to Cal.l.30 or Cal.f.4 treated plants, suggesting a strong synergistic effect against *B. cinerea* ingression and development in tomato leaves.

The ability of PGPM in sensitizing and priming the plant immune defense mainly relies on extracellular compounds synthesized and secreted by the beneficial microbes [54]. Recent studies have demonstrated that cell-free culture filtrates (CFCF) of ISR-inducing beneficial *B. amyloliquefaciens* MBI600, when inoculated by leaf spraying or root drenching, activated the expression of SA and/or JA defense-related genes [12]. The ISR-eliciting activity of CFCF was attributed to *Bacillus* sp.-secreted metabolites with antibiotic functions, such as iturinic-type lipopeptides, iturin A and bacilomycin D, as well as to the lipopeptides fengycin and surfactin [55]. In particular, it has been demonstrated that the spraying of *A. thaliana* leaves with iturin A or mucosubtilin activated the expression of SA and JA defense-related genes, *PR1* and *PDF1.2* [56], while *A. thaliana* seedling treatment with mycosubtilin from *B. subtilis* J-15 activated the expression of ET and JA defense-related genes [57]. Similarly, root treatment of tomato plants with fengycin stimulated the induction of SA-responsive genes [58]. In addition, treatment of mandarin fruit with each of the lipopeptides Iturin A, fengycin and surfactin secreted from *B. subtillis* ABS-S14 activated the expression of SA and JA defense-related genes [59], while treatment of tomato fruit with iturin A or bacilomycin D activated the expression of SA defense-related genes [60,61]. Previous integrated genomic and metabolomic analysis demonstrated that *B. halotolerans* strains Cal.l.30, Cal.f.4 and Hil4 have the ability to constitutively and simultaneously synthesize and secrete the lipopeptides fengycin, surfactin and the iturinic-type lipopeptide mojavensin [17,21], suggesting that colonization of plant tissues with these *B. halotolerans* strains may activate and sustained the defense-priming state of the host plant.

## 5. Conclusions

Beneficial microorganisms have been thought to trigger mild plant responses that do not affect the plant host defense-related transcriptome overall and contribute to sensitization rather than the actual induction of defense-related genes [7]. In the present study, *B. halotolerans* strains Cal.l.30 and Cal.f.4 belong to a special group of beneficial microorganisms that are able not only to sensitize the induction of defense-related genes upon pathogen challenge but also enable the host plant to maintain elevated levels of defense components around the threshold for effective resistance. Cal.l.30 and Cal.f.4 created a compatible consortium with both plant growth-promoting effects and direct biocontrol activity against *B. cinerea*. Furthermore, our findings highlighted that seed treatment followed by the soil drenching of tomato plants with the novel ISR-inducing *B. halotolerans* strains Cal.l.30 and Cal.f.4, as both individual strains and as a compatible consortium, could be an effective and practical approach for controlling gray mold disease under greenhouse crop production conditions. In this regard, the ability of *B. halotolerans* strains to promote growth and activate plants’ innate immune responses, along with their potential to synthesize numerous ISR-inducing and antifungal metabolites, make these beneficial *Bacillus* strains important for the development of effective biological control agents. 

## Figures and Tables

**Figure 1 biology-12-00779-f001:**
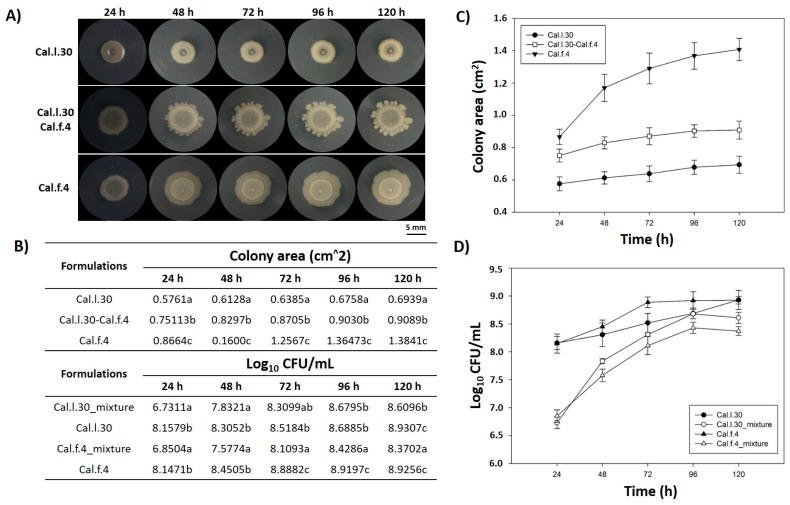
Mixed colony spot of *B. halotolerans* Cal.l.30 and *B. halotolerans* Cal.f.4 as a two-strain mixture on NA agar plates. (**A**) Colony expansion patterns derived from single- and two-strain communities were recorded at the indicated time intervals (24, 48, 72, 96 and 120 h). (**B**–**D**) Plots and tables of the average colony-covered area and number of bacterial cells versus time for the spot-inoculated two-strain and single-strain mixture. Graphs represent the mean ± SD from three independent experiments (*n* = 4). Statistical analysis was performed using one-way ANOVA followed by Tukey’s multiple comparison post hoc test (*p* < 0.05). Different letters indicate statistically significant differences among formulations at each time point.

**Figure 2 biology-12-00779-f002:**
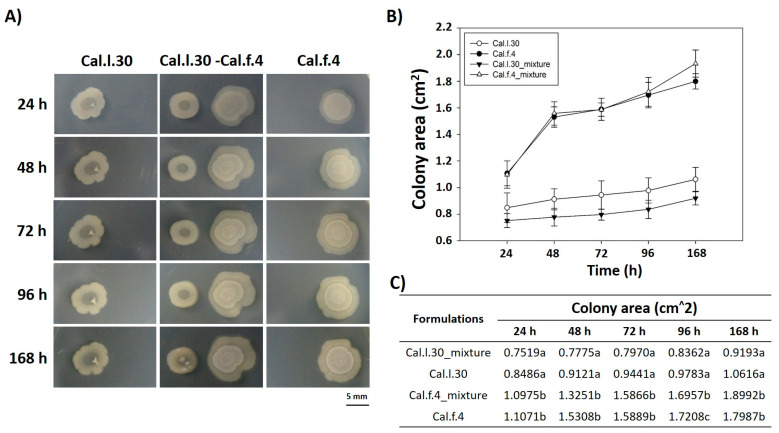
Time course of pairwise interactions between Cal.l.30 and Cal.f.4 bacterial colonies inoculated with a separation of 10 mm. (**A**) Colony expansion patterns derived from single- and two-strain communities were recorded at the indicated time intervals, 24 h, 48 h, 72 h, 96 h and 168 h. (**B**,**C**) Plot and table of the average colony-covered area versus time for the single strain and the interacting strains. Statistical analysis was performed using one-way ANOVA followed by Tukey’s multiple comparison post hoc test (*p* < 0.05). Different letters suggest statistical differences among treatments at each time point.

**Figure 3 biology-12-00779-f003:**
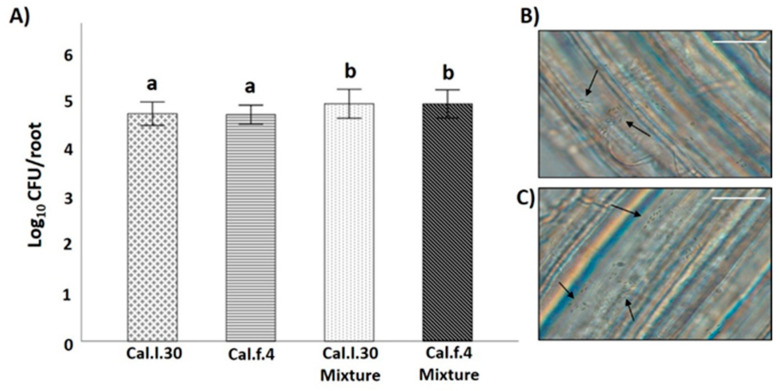
*B. halotolerans* Cal.l.30 and Cal.f.4 co-colonizing *A. thaliana* roots. (**A**) Colonization ability of the bacteria was measured as CFUs per root (*n* = 6). The samples were collected at 18 h. Statistical analysis was performed using one-way ANOVA followed by Tukey’s multiple comparison post hoc test (*p* < 0.05). Different letters suggest statistical differences among treatments. *A. thaliana* root colonization (black arrows) by (**B**) Cal.l.30 and (**C**) Cal.f.4 using an Olympus BX40 optical microscope (Scale bar: 20 nm).

**Figure 4 biology-12-00779-f004:**
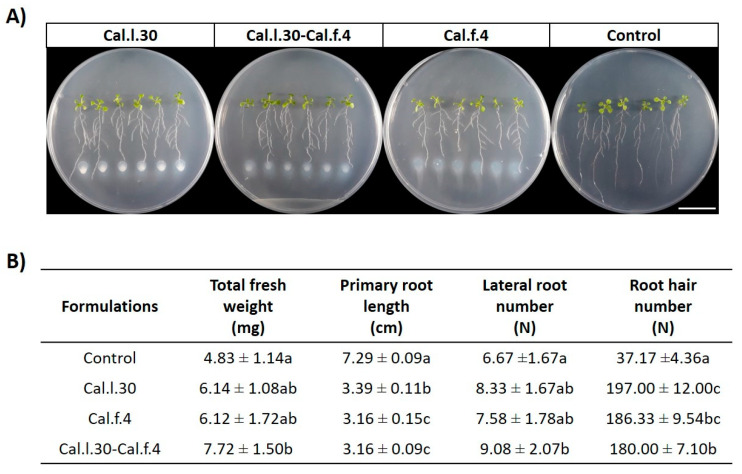
(**A**) Representative phenotypic responses of co-cultivation of *A. thaliana* seedlings with Cal.l.30, Cal.f.4 or Cal.l.30 + Cal.f.4 bacterial mixture inoculated at 3 cm distance from root tips. Physical changes were observed at 6 days after bacterial application (Scale bar: 20 mm). (**B**) Effect of endophytic bacterial strains Cal.l.30 and Cal.f.4 as mono- and co-cultures on different growth parameters (total fresh weight, primary root length, lateral root number and root hair number) of *A. thaliana* seedlings under in vitro conditions. Statistical analysis was performed using one-way ANOVA followed by Tukey’s multiple comparison post hoc test (*p* < 0.05). Different letters suggest statistical differences among treatments.

**Figure 5 biology-12-00779-f005:**
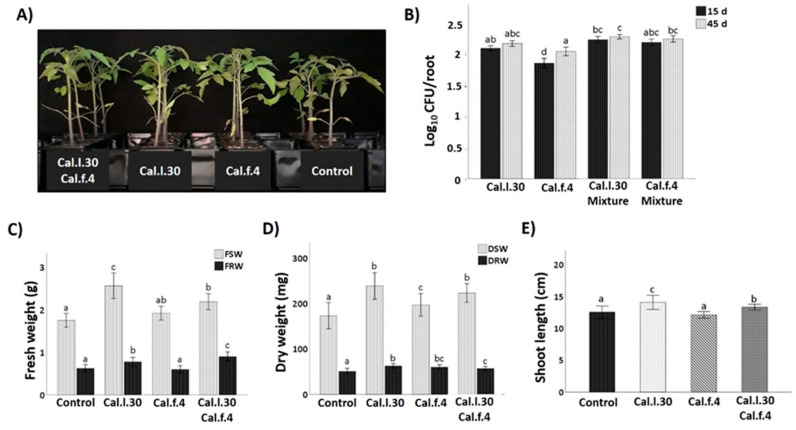
Effect of seed bio-priming with *B. halotolerans* strains, single or as mixture, on plant biomass accumulation in tomato at 45 days. (**A**) Representative image of the tomato plants inoculated with mono- and co-cultures of *B. halotolerans* strains at 45 days. (**B**) *B. halotolerans* populations of mono- and co-cultures colonizing tomato roots at 15 and 45 das. (**C**–**E**) The effect of different treatments on tomato plant height (shoot length), fresh shoot weight (FSW), fresh root weight (FRW), dry shoot weight (DSW) and dry root weight (DRW). Statistical analysis was performed using one-way ANOVA followed by Tukey’s multiple comparison post hoc test (*p* < 0.05). Different letters suggest statistical differences among treatments.

**Figure 6 biology-12-00779-f006:**
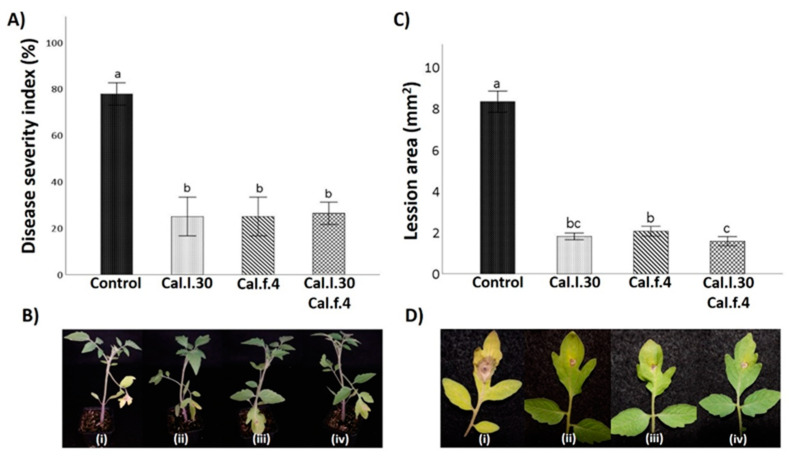
Effect of *B. halotolerans* strains Cal.l.30 and Cal.f.4 (singly or as mixture) on tomato leaf area covered with *B. cinerea* infection. (**B**–**D**) Representative images of *B. cinerea* symptoms development in leaves treated with Cal.l.30 (ii), Cal.f.4 (iii), Cal.l.30 + Cal.f.4 (iv) and control (i) tomato plants. (**A**) Disease severity index (%) and (**C**) Lesion area of infected tomato leaves was calculated. Values with a same letter are not significantly (*p* ≤ 0.05) different according to Turkey’s test. Numerical values were mean ± SD of ten replicates.

**Figure 7 biology-12-00779-f007:**
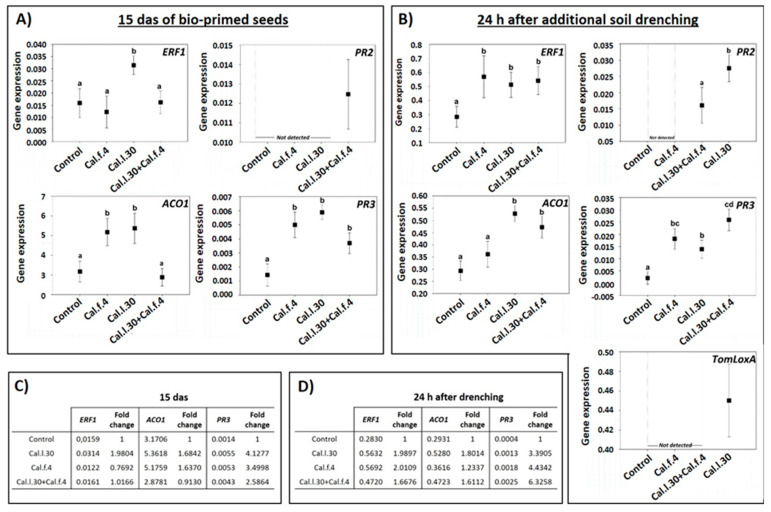
Expression of defense-related genes in leaves of young tomato plantlets after Cal.f.4, Cal.l.30 and Cal.l.30 + Cal.f.4 bacterial inoculation. (**A**) Gene expression of *ERF1*, *ACO1*, *PR2* and *PR3* 15 das of bio-primed tomato seeds. (**B**) Gene expression levels of *ERF1*, *ACO1*, *PR2*, *PR3* and *TomLoxA* 24 h after additional bacterial soil drenching. Graphs indicate the mean of gene expression for each treatment and a 95% confidence interval (Mean ± CI). Different letters suggest statistical differences among treatments after one-way ANOVA followed by Tukey’s multiple comparison post hoc test (*p* < 0.05). (**C**) Fold-change in gene expression of *ERF1*, *ACO1* and *PR3* 15 das of bio-primed tomato seeds. (**D**) Fold-change in gene expression of *ERF1*, *ACO1* and *PR3* 24 h after additional bacterial soil drenching.

**Figure 8 biology-12-00779-f008:**
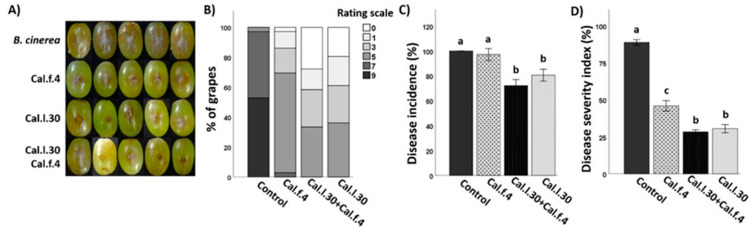
Inhibition of *B. cinerea* on wounded grape berries after Cal.l.30 and Cal.f.4 single strain and co-culture formulation. (**A**) Inhibition of gray mold disease after mono- and co-culture of Cal.l.30 and Cal.f.4 compared to control (*B. cinerea*) treatment. (**B**) Infected grapes (%) as assessed by color in control (*B. cinerea*) and inoculated grapes (*n* = 20). Colored blocks within each column represent the percentage of grapes corresponding to the scale value of the disease severity of *B. cinerea*. Visual disease rating scale of symptoms caused by *B. cinerea* is represented as 0 = healthy fruits, 1 = 1–10%, 3 = 11–25%, 5 = 26–50%, 7 = 51–75% and 9 ≥ 75% infected area. (**C**) Disease severity index (%) and (**D**) disease incidence (%) of wounded grapes, respectively. Bars represent the mean ± SD of 3 independent biological replicates. Letters indicate statistically significant differences among treatments after Tukey analysis (*p* < 0.05).

## Data Availability

The *Bacillus halotolerans* strains Cal.l.30 and Cal.f.4 whole genome projects are available in the NCBI database under the accession numbers JAEACK000000000, JAEACI000000000 (GenBank), respectively, and SAMN16949411, SAMN16949417, (BioSample) and PRJNA681331, PRJNA681333, (BioProject), respectively.

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
