# Peer review of "Compatible Consortium of Endophytic Bacillus halotolerans Strains Cal.l.30 and Cal.f.4 Promotes Plant Growth and Induces Systemic Resistance against Botrytis cinerea"

_biology, 2023, doi:10.3390/biology12060779_

Round 1

Reviewer 1 Report

The manuscript is sufficiently novel and interesting and the research questions are relevant and important. The manuscript is clearly laid out and all the key elements are present.

The manuscript presents an interesting topic about possibilities of applying microorganisms in agricultural production. Numerous studies are focused towards reducing the use of chemical fertilizers and pesticides and their replacement with eco-friendly and sustainable strategies. Application of microorganisms is indispensable component of sustainable agricultural and in the organic food production. Examination of different plant growth promoting characteristics of the microorganism is an important aspect in the selection of microorganisms for microbial fertilizers and pesticides. The manuscript presents an effective and practical approach for controlling fungal and bacterial diseases and enhances plant growth.

Introduction clearly describes the problem being investigated, summarizes relevant research, explains other authors' findings and describes the aim of experiment

The experiment design is suitable for answering the question posed and the author accurately explains how the data was collected, and materials and experimental techniques have been adequately and correctly described.

Appropriate analysis of the results was carried out, and clearly presented in the tables and figures. The authors explained results clearly and properly compared with the results of other authors. Discussion supports obtained results. The authors indicate that the results are related and consistent with the expected and previous researches and support previous theories.

In conclusion, the authors point out the practical significance and the potential application of the results in sustainable agricultural production.

However, these are some suggestions for improvement.

In the Material and Methods section (section 2.4.1, lines 185-190):

“The pot experiment was conducted in a greenhouse (average air temperature 25°C, relative air humidity 60%)…” – What was the daily photoperiod (hours light and hours dark) for the entire experiment?

In how many replications the experiment was set up?

How many plants were per pot?

In which substrate the plants were grown? (Soil, mixture of soil and sand or some other).

Reviewer 2 Report

The article “Comparative efficacy of endophytic Bacillus halotolerans strains Cal.l.30 and Cal.f.4 used singly or as a mixture to promote plant growth, induce systemic resistance and mediate biocontrol activity against Botrytis cinerea ” is devoted to the important and acute theme of the influence of microbial consortiums on plant growth and resistance. The authors really did a great job and provided enough evidence of the possibility of combined application of Cal.l.30 and Cal.f.4. The article is well-structured and detailed.

I have some minor comments:

1)Title: Now it's very long. May be “Consortium of endophytic Bacillus halotolerans strains Cal.l.30 and Cal.f.4 possesses plant growth promote activity and induces systemic resistance against Botrytis cinerea”

2)Introduction: Please, give 2-3 sentences about strains (for example, insert lines 111-112). Why were these strains used?

3)Line 119: Describe Cal.1.30-rif.

4)In Fig 1 Graphs C,D duplicate data in table B. In my opinion, Table B provides a much more clear idea on the topic under consideration. In Fig 2 B duplicates table C.

5)In Fig 8 the sequence of variants should be corrected (it must be as on other figures).

6)Conclusions: Please, give a clear numbered list of conclusions on your work.

Reviewer 3 Report

Simple summary:

- Line 22: 'in vitro' in italics

Introduction:

- Line 48-49: Reiteration. Be careful, this happens some other time along the introduction section

- Be careful, first time you use the abbreviations you need to write the full name

- In some moments seems more a discussion (IRA paragraph)

- General: Seems very short for the topics included, I think it's improvable

Materials and Methods

- rpm of culture is necessary

- Line 119: Cal.l.30-rif? New strain? Antibiotic resistant? You explain later, but this is not the right order to place the statements, but, why only one of the strains is resistant? Again, explained later, untidy...

- 10^8 CFU used to be a little more than 0.5 of OD600. Maybe you need to include this is based in your own growth curve

- The compatibility test seems not very strong... Or just maybe not well described, but don't give me much confidence

- Explain surface sterilization of seeds

- Coming back to rif: I don't think only with one resistant strain is enough to get assumptions from both... I know you specify some of the results for the rif resistant, but it created confusion. Not the best design

- Need to describe greenhouse conditions: soil, temperature, climatic conditions...

- Description about drenching inoculum....

- How to determine colonization...again, too short to estimate the other strain

- Line 243: careful, some orthographic mistakes

- What? Now you changed to berries? Not sure this design is ok, but as a complementary test may work...

Results:

- Not fully sure of coexistence results....

- Pictures without scale... not valid

- Figure 2: I'm not agreeing with your conclusions after seen these colonies, seem smaller, but again, not scale...

- I cannot back the results of co-inoculantion in vitro... Not even close to a strong result....

- Combination of both strains is not clearly showing a good results, at least better than individual... barely trustable in respect to control...

- Again, scales....

- Biocontrol seems solid, however, again the combination doesn't seem with a better effect

- figure 8: Light on grapes seems not to be the same, I cannot back these results

Discussion:

- Quite decent, but a good deal of relevant literature on the topic that is not included

Conclusions:

- Some parts of this section are not conclusions, please avoid

- Citations in the conclusions are not ok

- Be careful, this is not a resume, it is a conclusions section

Pretty ok

Round 2

Reviewer 3 Report

Still not agree with the design, specially with the rif mutants and morphological identification. I understand the point, but it's not the right way to do it and may create inconsistencies. Moreover, other approaches are not the best way as well. I leave this in the editor hands because I fell the results are good and interesting, but the way to approach is not the right one. I cannot validate this under the shape it is presented. Furthermore, I can understand that it may be difficult for authors to repeat tests or to incorporate some others, but the present work is not something I can support as it is, is not solid enough. I'0m sorry

ok